# Functional Characterization of *Physcomitrella*
*patens* Glycerol-3-Phosphate Acyltransferase 9 and an Increase in Seed Oil Content in Arabidopsis by Its Ectopic Expression

**DOI:** 10.3390/plants8080284

**Published:** 2019-08-13

**Authors:** Sun Ui Yang, Juyoung Kim, Hyojin Kim, Mi Chung Suh

**Affiliations:** 1Department of Bioenergy Science and Technology, Chonnam National University, Gwangju 61186, Korea; 2Department of Life Science, Sogang University, Seoul 04107, Korea

**Keywords:** arabidopsis, glycerol-3-phosphate acyltransferase, GPAT9, polyunsaturated fatty acids, *Physcomitrella patens*, storage oil, triacylglycerol

## Abstract

Since vegetable oils (usually triacylglycerol [TAG]) are extensively used as food and raw materials, an increase in storage oil content and production of valuable polyunsaturated fatty acids (PUFAs) in transgenic plants is desirable. In this study, a gene encoding glycerol-3-phosphate acyltransferase 9 (GPAT9), which catalyzes the synthesis of lysophosphatidic acid (LPA) from a glycerol-3-phosphate and acyl-CoA, was isolated from *Physcomitrella patens*, which produces high levels of very-long-chain PUFAs in protonema and gametophores. *P. patens* GPAT9 shares approximately 50%, 60%, and 70% amino acid similarity with GPAT9 from *Chlamydomonas reinhardtii*, *Klebsormidium nitens*, and *Arabidopsis thaliana*, respectively. *PpGPAT9* transcripts were detected in both the protonema and gametophores. Fluorescent signals from the *eYFP:PpGPAT9* construct were observed in the ER of *Nicotiana benthamiana* leaf epidermal cells. Ectopic expression of *PpGPAT9* increased the seed oil content by approximately 10% in Arabidopsis. The levels of PUFAs (18:2, 18:3, and 20:2) and saturated FAs (16:0, 18:0, and 20:0) increased by 60% and 43%, respectively, in the storage oil of the transgenic seeds when compared with the wild type. The transgenic embryos with increased oil content contained larger embryonic cells than the wild type. Thus, *PpGPAT9* may be a novel genetic resource to enhance storage oil yields from oilseed crops.

## 1. Introduction

In plants, triacylglycerol (TAG) is an energy-rich storage compound in which three fatty acid molecules are esterified to glycerol [1,2]. TAGs obtained mainly from seeds and fruits are used as edible oils, such as frying oils, and salad dressing or as raw materials to produce detergents, lubricants, paints, linoleum, coatings, and biodiesel [2,3]. In addition, global warming and depletion of fossil fuels have led to increased production of vegetable oils as alternative and sustainable fuels and biomaterial resources [4,5,6]. In particular, the production of vegetable oils with enhanced polyunsaturated fatty acids (PUFAs), which are essential for human health and nutrition, has been a major area of focus in the field of plant biotechnology [7,8].

De novo fatty acid biosynthesis occurs exclusively in plastids, and glycerolipids are synthesized via prokaryotic pathway in the plastids and eukaryotic pathway in the endoplasmic reticulum (ER) [9]. The synthesized fatty acyl-acyl carrier protein (ACP) or acyl-CoA is esterified to glycerol-3-phosphate by glycerol-3-phosphate acyltransferase (GPAT) [10,11,12]. The produced lysophosphatidic acid is subsequently converted to phosphatidic acid (PA) via acylation by lysophosphatidic acid acyltransferase [13]. In plastids, PA is used for the synthesis of phosphatidyl glycerol (PG) or dephosphorylated to diacylglycerol (DAG), which is the precursor of galactolipid and sulfolipid syntheses, by PA phosphatase (PAP) [14]. The 16:0 and 18:1 fatty acids on PG, monogalactosyl diacylglycerol (MGDG), and digalactosyl diacylglycerol (DGDG) are desaturated by fatty acid desaturase (FAD)4, FAD5, FAD6, FAD7, or FAD8 [15,16,17,18,19]. In the ER, PA, which is the common precursor for the synthesis of phospholipids, is converted to DAG by PP for the synthesis of phosphatidylcholine (PC), phosphatidylethanolamine (PE), and phosphatidylserine (PS) or cytidine diphosphate (CDP)-DAG by CDP-diacylglycerol synthetase for the synthesis of phosphatidylinositol (PI) [14,20,21]. DAG is acylated by acyl-CoA to TAG by diacylglycerol acyltransferase (DGAT) [22]. Phospholipid:diacylglycerol acyltransferase (PDAT) also contributes TAG biosynthesis by transferring the acyl-CoA from PC to DAG [23]. The C18:1 fatty acid on PC is further desaturated to C18:2 and C18:3 by FAD2 and FAD3 [24,25], respectively, or C16:0- and C18:0-CoAs are further elongated to very long chain FAs by the fatty acid elongase complex, which is composed of β-ketoacyl-CoA synthase, β-ketoacyl-CoA reductase, β-hydroxy acyl-CoA dehydratase, and enoyl-CoA reductase [26].

Since the activity of GPAT, which catalyzes the committed step in the synthesis of glycerolipids, has been identified in liver cell extracts of the guinea pig (*Cavia porcellus*) [27] and microsomal fraction of avocado (*Persea americana*) mesocarps [28], the presence of GPATs has been reported in various organisms, including bacteria, fungi, diatoms, and plants [29,30,31,32]. Recently, in vivo functions of GPAT9, which is the homolog of a microsomal GPAT involved in the production of storage oil in mammalian cells, were characterized in Arabidopsis [11]. Loss-of-function mutant *atgpat9* displayed a lethal embryo phenotype, suggesting the essential role of GPAT9 in membrane lipid synthesis [11]. Total seed oil content was reduced by 26 to 44% in seed-specific *GPAT9* knockdown Arabidopsis plants generated by the expression of artificial microRNA of *GPAT9,* indicating that GPAT9 is involved in TAG synthesis in Arabidopsis seeds [11]. In addition, AtGPAT9 exhibits sn-1 acyltransferase activity with preferential specificity for acyl-CoAs in the microsomal fractions prepared from yeast cells expressing *AtGPAT9* [33]. Recently, GPAT9-type GPAT overexpression greatly enhances TAG accumulation in red alga (*Cyanidioschyzon merolae*) [34]. Overexpression of *AtGPAT9* increased polar lipid and TAG levels in the leaves and enhanced lipid droplet accumulation in the pollen grains of Arabidopsis [33], suggesting that GPAT9 can be a novel genetic resource for the enhancement of storage oil yield from seeds or vegetative tissues.

The moss *Physcomitrella patens* is one of the earliest land plants, and it is a lower plant model. In the life cycle of *P. patens*, the haploid spore forms filamentous protonema with two distinct chloronema and caulonema cell types. The protonema forms buds, which are able to generate gametophores, shoot-like stems bearing phylloids, and rhizoids [35,36]. Female and male sex organs are formed at the tip of the gametophore, motile spermatozoids are fertilized with the egg cell, and then the diploid sporophyte develops from the zygote. The haploid spores are generated by meiosis in an apical spore capsule of the sporophyte. In particular, *P. patens* protonema and gametophore possess approximately 65% long-chain (LC)-PUFAs and very-long-chain (VLC)-PUFAs, indicating that characterization of genes encoding enzymes involved in LC- and VLC-PUFA biosyntheses may be important in the production of PUFAs in transgenic plants [36,37,38].

In this study, a gene encoding GPAT9, which catalyzes the first step in the synthesis of membrane lipids and TAG, was isolated from *P. patens* by using homology searches with the deduced amino acid sequences of Arabidopsis *GPAT9*. The deduced amino acid sequences of *GPAT9* from *P. patens* and other plant species were compared to examine the genetic diversity of *GPAT* in plants. To investigate the role of the isolated *PpGPAT9*, its mRNA expression pattern in the gametophore and protonema and subcellular localization of eYFP:PpGPAT9 were examined. An increase in the size of the seeds and embryonic cells and in seed mass was observed in transgenic Arabidopsis lines ectopically expressing *PpGPAT9* driven by the *CaMV 35S* promoter. Finally, ectopic expression of *PpGPAT9* increased the total seed oil content, with significantly higher levels of PUFAs in the transgenic Arabidopsis. These results indicate that *PpGPAT9* may be used for the enhanced production of storage oils in transgenic oilseed crops.

## 2. Results

### 2.1. Isolation of GPAT9 From P. patens

BLAST was used to isolate *GPAT9* from *P. patens* with the deduced amino acid sequence of Arabidopsis *GPAT9*, which is functionally characterized [11,33] at the Phytozome v12.1 public database (https://www.phytozome.net; e-value ≤ 1 × 10^−5^). Ten genes encoding GPAT were present in the genome of *P. patens*. When a phylogenetic tree of Arabidopsis and *P. patens GPATs* was constructed (Figure 1A), plastidial soluble *GPAT/ATS1* was initially classified with the other membrane-bound *GPAT* groups*. GPAT9*, involved in membrane lipid and TAG biosyntheses, was further classified with other *GPATs*, including *GPATs* required for the synthesis of cutin (*AtGPAT4, AtGPAT8*, and *AtGPAT6*), suberin (*AtGPAT5* and *AtGPAT7*), and functions in the mitochondria (*AtGPAT1*) [11,33,39,40,41,42,43].

In addition, *P. patens* GPAT9 homologues were isolated from algae (*Lobosphaera incisa*, *Chlamydomonas reinhardtii*, and *Klebsormidium nitens*) and lower (*Selaginella moellendorffii*) and higher (*Amborella trichopoda*, *Solanum lycopersicum*, *Arabidopsis thaliana*, *Brachypodium distachyon*, and *Oryza sativa*) plants to examine the genetic diversity of GPAT9. In a phylogenetic analysis of PpGPAT and other GPAT homologues (Figure 1B), GPAT9 is evolutionarily conserved and a single-copy gene in the examined plant species. The four conserved motifs (BOX I to IV) in GPAT9 are shown in Figure 1C, and the amino acid residues, His and Asp in BOX I, Gly in BOX III, and Pro in BOX IV, which are essential for the *E. coli* GPAT enzyme activity assay [44], are highly conserved in plant GPAT9s. Although four amino acid residues, Phe and Arg in BOX II and Glu and Ser in BOX III, are involved in the binding of the glycerol-3-phosphate substrate in *E. coli* [44], only two, Arg and Glu, residues were conserved in plant GPAT9s (Figure 1C).

### 2.2. Expression of PpGPAT9 in the Protonema and Gametophores

To investigate the expression patterns of *PpGPAT9* during the life cycle of *P. patens*, the expression of *PpGPAT9* was analyzed in the Physcomitrella eFP browser (http://bar.utoronto.ca/efp_physcomitrella/cgi-bin/efpWeb.cgi). When the relative expression of *PpGPAT9* (*Pp1s138_27V6.1*, *Phpat.011G104300*) vs. *PpActin2* (*PpACT2*, *Pp1s198_157V6.1*, *Phpat.003G133100*) was calculated, the highest expression of *PpGPAT9* was detected in the mature sporophyte (SM), followed by the spores. The expression of *PpGPAT9* relative to *PpActin2* was approximately 1.2-, 1.3-, 1.4-, and 1.4-fold higher in the rhizoids, protoplasts, gametophores, and S1 sporophyte stage, but approximately 0.6- to 0.8-fold lower in the protonema, archegonia, and S2 and S3 sporophyte stages (Figure 2A); this indicated that *PpGPAT9* was ubiquitously expressed in all the tested *P. patens* organs. To examine *PpGPAT9* transcript levels, quantitative RT-PCR was performed using 7-day-old protonema and 4- and 8-week-old gametophores (Figure 2B). The PpGPAT9 transcript levels were approximately 3-fold and 1.6-fold higher in 4-week-old and 8-week-old gametophores than in the protonema (Figure 2C).

### 2.3. Subcellular Localization of PpGPAT9

To examine the subcellular localization of PpGPAT9, we constructed a binary vector, *peYFP:PpGPAT9*, in which the enhanced yellow fluorescent protein (eYFP) was fused to the 5′-end of *PpGPAT9* under the control of the *CaMV 35S* promoter (Figure 3A). The generated construct was co-transformed into *Nicotiana benthamiana* leaves with the *BrFAD2-1:mRFP* construct, which is localized in the ER [45], via *Agrobacterium*-mediated infiltration. The yellow fluorescent signals from *peYFP:PpGPAT9* were merged with the red fluorescent signals from *BrFAD2-1:mRFP* (Figure 3B). In the topology analysis of PpGPAT9 by using an open-source tool for visualization of proteoforms (http://topcons.net/; http://wlab.ethz.ch/protter/start/), PpGPAT9 was predicted to possess an N-terminal region oriented to the ER lumen and one ER lumen loop, three transmembrane domains, and a long C-terminal cytoplasmic region (Figure 3C). Four acyltransferase motifs (BOX I to BOX IV) and an ER retrieval motif (“VIRRL” residues) [46] were observed to be present in the long C-terminal cytoplasmic regions. Therefore, subcellular localization and topology analysis of PpGPAT9 indicate that *PpGPAT9* is localized in the ER and may be involved in eukaryotic pathway of glycerolipid synthesis in the ER.

### 2.4. Generation of Arabidopsis Transgenic Plants Ectopically Expressing PpGPAT9

To examine the in vivo activity of PpGPAT9, a pBA002 binary vector construct, in which *PpGPAT9* was translationally fused to the 3′-end of 6x MYC under the control of the *CaMV 35S* promoter was generated, and then introduced to Arabidopsis plants via *Agrobacterium*-mediated dip floral method (Figure 4A). Transgenic Arabidopsis seeds were selected from half-strength MS medium supplemented with 10 μg/mL phosphinothricin (PPT). Chromosomal DNA was extracted from the leaves of PPT-resistant plants, and PCR was performed with a gene-specific primer set (PpGPAT9 BamH1 F1 and PpgPAT9 Spe1 R1; Appendix A) to ensure the presence of the transferred DNA (T-DNA) region in the Arabidopsis chromosome. Figure 4B shows that the PCR bands were detected in three independent transgenic lines (#2, #7, and #13) but not in the wild type, indicating that the T-DNA region harboring *PpGPAT9* is integrated into the Arabidopsis chromosome. To examine the expression of PpGPAT9 in the transgenic Arabidopsis lines, total proteins were extracted from #2, #7, and #13 seedlings (T_2_ generation); separated using 10% SDS-polyacrylamide gel electrophoresis (PAGE); and then visualized using anti-c-Myc antibody and an ECL chemiluminescence detection system. Bands corresponding to approximately 57 kDa were clearly detected in three transgenic lines, but not in the wild type (Figure 4C), indicating that the PpGPAT9 proteins were expressed in the transgenic Arabidopsis plants.

### 2.5. Morphological and Microscopic Analyses of Transgenic Arabidopsis Seeds Ectopically Expressing PpGPAT9

During the growth and development of transgenic Arabidopsis plants ectopically expressing *PpGPAT9* (*PpGPAT9* OX), no noticeable differences were observed relative to the wild type. We next examined whether the ectopic expression of *PpGPAT9* affects morphological changes in the *PpGPAT9* OX seeds. When the size and weight of seeds from the wild type and three *PpGPAT9* OX lines were measured, the size of seeds from the #2, #7, and #13 transgenic plants increased by approximately 6%, 6%, and 16%, respectively, compared with the wild type (Figure 5A). We also observed that the weight of seeds from the #2, #7, and #13 transgenic plants was approximately 3%, 2%, and 6% higher, respectively, than that from the wild type (Figure 5B).

We subsequently analyzed if the increase in *PpGPAT9* OX seed size is associated with cell division or expansion. After the wild type and *PpGPAT9* OX seeds were cleared using Visikol, the size of the cotyledons and embryonic cells was measured under a microscope ICC50 HD (Leica, Wetzlar, Germany). The size of #2, #7, and #13 *PpGPAT9* OX cotyledons was approximately 13, 18, and 15% larger than that of the wild type (Figure 5C,D). The size of #2, #7, and #13 *PpGPAT9* OX embryonic cells also increased by approximately 10%, 7%, and 8%, respectively, when compared with the wild type (Figure 5C,D). When the number of embryonic cells in the wild type and transgenic cotyledons was calculated, significant differences were not observed (Figure 5E). These results suggest that the increased seed size of *PpGPAT9* OX lines may be attributable to cell expansion, but not cell division.

### 2.6. Fatty Acid Analysis of Arabidopsis Seeds and Leaves Ectopically Expressing PpGPAT9

To examine the fatty acid composition and levels in the Arabidopsis seeds and leaves ectopically expressing *PpGPAT9*, total fatty acids were extracted from dry seeds and leaves of 4 week-old-plants and subsequently analyzed using GC-FID. The levels of total fatty acid methyl esters (FAMEs), which are mainly from TAGs [48,49,50], increased by approximately 10% in the seeds of three transgenic lines (#2, #7, and #13) relative to the wild type (Figure 6A). The levels of both saturated (16:0, 18:0, and 20:0) and polyunsaturated (18:2, 18:3, and 20:2) fatty acids increased by approximately 44% and 60%, whereas the levels of monounsaturated FAs (C18:1 and C20:1) were not significantly altered or C22:1 levels decreased in all the transgenic seeds when compared with the wild type (Figure 6C). The largest increase (approximately 53%) was detected in the levels of C18:3 plus C20:2 fatty acids in the transgenic seeds.

In *PpGPAT9* OX leaves relative to the wild type, the levels of C16:1, C18:0, and C20:2 increased by approximately 2 to 9%, 11%, and 24%, respectively, but no significant changes were observed in the levels of total fatty acids, which are mainly from membrane lipids [48,49,50] (Figure 6B,D). No noticeable accumulation of oil bodies was observed during the transient expression of *PpGPAT9* in *N. benthamiana* leaves (Figure 7); under similar conditions, oil bodies were clearly detected by the transient expression of Arabidopsis *WRINKLED1* [51]. These results indicate that PpGPAT9 functions in glycerolipid synthesis and may perform preferential acylation of LC- and VLC-PUFAs to glycerol-3-phosphate.

## 3. Discussion

The lower plant *P. patens* harbors relatively higher amounts of LC-PUFAs and VLC-PUFAs such as C18:2, C18:3, and C20:4 in cellular membranes [36,37,52]. However, there is limited information on the activity and substrate specificity of acyltransferases required for the synthesis of glycerolipids in *P. patens*. Here, we have reported that (1) the expression of *PpGPAT9* in the protonema and gametophores was detected using qRT-PCR. (2) PpGPAT9 is localized in the ER. (3) Ectopic expression of *PpGPAT9* increased the size of the seeds and embryonic cells, which is related to the increased cell expansion. (4) A significant increase in the levels of total FAMEs was observed in *PpGPAT9* OX seeds relative to the wild type, but not in *PpGPAT9* OX leaves. (5) An increase in LC-FA and VLC-FA levels relative to saturated FA levels was prominent in the *PpGPAT9* OX seeds and leaves. Our current study shows that PpGPAT9 is a novel genetic material that can be used for the increased production of storage oils with more PUFAs in transgenic oilseed plants.

Genome-wide analysis has shown that multi-copy *GPAT* genes are present in land plants. Arabidopsis harbors 10 GPATs, which function in the ER (GPAT4-9, ER membrane-bound form), mitochondria (GPAT1-3, membrane-bound form), and chloroplast (ATS1, soluble form) [11,42,43,46,53]. All GPATs, including PpGPAT9, contain highly conserved acyltransferase domains, which include conserved amino acid residues, His and Asp in BOX I, Arg in BOX II, Glu and Gly in BOX III, and Pro in BOX IV. GPAT4, GPAT6, and GPAT8 have additional conserved domains (the first Asp in motif I, the first Lys in motif III, and the first Asp in motif IV), which are essential for the phosphatase activity that enables the production of 2-monoacylglycerol, a precursor for cutin synthesis [42,54]. In contrast, other GPATs, including GPAT5 and GPAT7 grouped in the suberin-biosynthesis clade (Figure 1A), have lost some amino acid residues at the active site required for phosphatase activity [37,41], indicating the functional divergence of the *GPAT* family.

*GPAT9* is present as a single-copy gene in most organisms, including *P. patens* (Figure 1) [46]. The Arabidopsis *gpat9* mutant displayed the lethal-embryo phenotype [11], indicating that GPAT9 is essential in Arabidopsis. In this study, we observed the ubiquitous expression of *PpGPAT9* during the life cycle of *P. patens* life (Figure 2). Interestingly, we tried to generate *P. patens gpat9* mutants by homologous recombination [35]; however, it was not successful under similar conditions in which knockout mutants of *PpGPAT2* and *PpGPAT4* involved in cutin synthesis were generated [55]. When *PpGPAT9* is overexpressed in Arabidopsis, the total FAME levels increased in *PpGPAT9* OX seeds, and alterations in the fatty acid composition of *PpGPAT9* OX seeds and leaves were observed when compared with the wild type (Figure 6). Therefore, these results indicate that PpGPAT9 is involved in glycerolipid synthesis and PpGPAT9, like AtGPAT9, may have an essential role in *P. patens*.

Gidda et al. [46] reported that Arabidopsis GPAT9 is localized to the ER in tobacco BY-2 suspension cells. A hydrophobic pentapeptide motif (-φ-X-X-K/R/D/E-φ-, φ; large hydrophobic amino acid residues) is essential for the ER retrieval of AtGPAT9 and AtFAD2 [46,56]. We also observed a similar result; the yellow fluorescent signals from the eYFP:PpGPAT9 construct were merged with the red fluorescent signals from the BrFAD2-1:mRFP construct in the ER of *N. benthamiana* leaves (Figure 3). Therefore, the “VIRRL” motif in the C-terminal region of PpGPAT9, which corresponds to the hydrophobic pentapeptide motif of AtGPAT9 and BrFAD2-1 [45,46], may be the ER retention signal of PpGPAT9. Alignment of the deduced amino acid sequences of lower and higher plant GPAT9s showed that the hydrophobic pentapeptide motif has been conserved from charophyte to monocot, except chlorophyte, during plant evolution. In the topology prediction analysis of PpGPAT9, we observed that PpGPAT9 harbors three membrane-spanning domains and its N-terminal and C-terminal regions were directed to the ER lumen and cytoplasm, respectively (Figure 3A), although both N- and C- terminal residues of AtGPAT9 were predicted to be oriented to the cytosol [46]. However, the catalytic domains of both AtGPAT9 and PpGPAT9 were observed to be present in the cytoplasm.

There have been attempts to increase storage oil yield by metabolic flux control during TAG biosynthesis. Vanhercke et al. [57] reported three critical steps: (1) The “PUSH” step to increase the levels of fatty acid precursors required for TAG synthesis, (2) “PULL” step to enhance TAG assembly, and (3) “PROTECT” step to prevent the degradation of synthesized TAG. In the “PUSH” step, overexpression of transcription factors such as WRI1 and LEC led to an increase in seed oils by approximately more than 40% [48,58,59,60,61]. In the “PULL” step, the oil content was increased by up to 28% in the transgenic Arabidopsis seeds expressing *DGAT1* or *PDAT1* [62,63]. In the “PROTECT” step, the inhibition of SDP1 lipase, which degrades TAG, resulted in increased oil content by up to 8% in the seeds or vegetative tissues of the transgenic Arabidopsis lines [64,65]. In particular, previous studies have reported that *GPAT9* genes of Arabidopsis and the marine diatom *Phaeodactylum tricornutum* also contribute to elevated TAG content in the transgenic Arabidopsis plants and *P. tricornutum*, respectively [29,33]. Ectopic expression of *PpGPAT9*, which shares approximately 70% amino acid sequence similarity with AtGPAT9, led to increased TAG production in the transgenic Arabidopsis seeds, confirming that *GPAT9* as well as *DGAT* and *PDAT* can be genetic resources involved in the “PULL” mechanism to increase the oil yield in a sustainable manner.

According to previous studies [48,57,60,66,67], seed mass and oil content increased in the transgenic lines expressing Arabidopsis or *Camelina* (*Camelina sativa*) *DGAT1* or Arabidopsis or rapeseed (*Brassica napus*) *WRINKLED1* (*WRI1*). In *C. sativa* cotyledons overexpressing *AtWRI1*, the increase in the size of embryonic cells was closely related to an increase in embryonic cell expansion rather than cell division [48]. However, both cell expansion and division increased in transgenic *C. sativa* cotyledons overexpressing *CsDGAT1B* [66]. In the current study, increased TAG levels in *PpGPAT9* OX seeds were observed to be proportional to the increased cotyledon size, which is closely related to the increased embryonic cell expansion rather than embryonic cell division (Figure 5). Weber et al. [68] reported that a high ratio of hexoses to sucrose activates embryonic cell growth via cell division, whereas a low ratio of hexoses to sucrose promotes differentiation and storage product synthesis. Therefore, the altered ratio of carbohydrate metabolites may be related with the cotyledon size in *PpGPAT9* OX seeds. However, the molecular networks between embryonic cell expansion and seed oil yield during seed development need to be studied further.

In the in vitro enzyme activity assay for sunflower (*Helianthus annuus*) GPAT9-1 by using yeast microsomal fractions, the substrate activity of HaGPAT9-1 was the strongest for C16:0-CoA, followed by C18:2-CoA and C18:1-CoA [69]. The result was consistent with the increased TAG species with C16:0 in yeast cells expressing *HaGPAT9* [69]. AtGPAT9 showed the highest substrate preference for C18:1-CoA relative to C16:0-CoA, C18:0-CoA, and C18:3-CoA, which is correlated with elevated C18:1 levels in the storage oils of transgenic seeds ectopically expressing *AtGPAT9* [35]. When *PtGPAT9* was expressed in *P. tricornutum*, a remarkable increase in PUFA levels and concomitant decrease in saturated FA and monounsaturated FA levels were observed. In particular, a significant increase (~40%) in the levels of C20:5, which are abundant (~18% per dry weight) fatty acids in *P. tricornutum*, and decrease (~45%) in C16:0 levels were detected in *PtGPAT9* overexpressing *P. tricornutum* when compared with the wild type [33]. Resemann et al. [38] reported that approximately 70% and 67% PUFAs are present in *P. patens* protonema and gametophores, respectively. The major PUFAs were C16:2, C16:3, C18:3, C20:4, and C20:5 in the protonema and C18:2, C18:3, and C20:4 in the gametophore [38]. In this study, the levels of C18:3 and C20:2 FAs were higher than that of saturated FAs in the transgenic seeds ectopically expressing *PpGPAT9* (Figure 6). It is interesting that ectopic expression of *PpGPAT9* caused a significant increase in C20:2 content, which is not the major component in Arabidopsis seeds. Overall, these results suggest that GPAT9 may exhibit a substrate preference for certain FAs with specific carbon chain length and/or specific degree of unsaturation and can be used for the improvement of oil quality in oilseed crops. In conclusion, this study shows that ER-localized PpGPAT9 is involved in glycerolipid synthesis and is a genetic resource that can be used to increase the production of storage oils with more PUFAs in oilseed crops.

## 4. Materials and Methods

### 4.1. Plant Materials and Growth Conditions

The wild type *P. patens* (Gransden 2007 strain) was obtained from Jeong Sheop Shin at Korea University and grown on BCD medium supplemented with 5 mM diammonium tartrate and 1 mM CaCl_2_, according to previous studies [70,71]. *A. thaliana* (ecotype Columbia-0, Col-0) and transgenic Arabidopsis plants were grown under moss growth conditions. Seeds were sterilized with 75% ethanol containing 0.05% Triton X-100 and then washed with 100% ethanol. Germinated 7-day-old seedlings grown on half-strength MS media containing 1% sucrose and 0.7% phytoagar were transplanted to an autoclaved soil mixture containing soil, vermiculite, and pearlite (3:2:1 v/v/v). The photoperiod was 16 h day and 8 h dark, and light intensity was 100–120 µmol/m/s at 23 °C.

### 4.2. Phylogenetic Tree Construction

Amino acid sequences of GPAT9 homolog genes from *P. patens, A. thaliana*, *C. reinhardtii*, *L. incisa*, *K. nitens*, *S. moellendorffii*, *A. trichopoda*, *B. distachyon*, *O. sativa*, and *S. lycopersicum* were obtained using BLAST with cutting threshold 1 × 10^−5^ from Phytozome v12.1 (www.phytozome.net), NCBI (www.ncbi.nlm.nih.gov), and UniProt (www.uniprot.org). The amino acid sequences were used for domain and similarity analyses with CLUSTALW software. The phylogenetic tree was created using the MEGA6 program with the Maximum Likelihood method [72].

### 4.3. RNA Isolation and Gene Expression Analysis

To isolate the total RNA from 7-day-old protonema and 4- and 8-week-old gametophores, the tissues were ground in liquid nitrogen and the total RNA was extracted using TRIzol reagent (Ambion, CA, USA), according to the manufacturer’s instructions. Two micrograms of the isolated total RNA was converted to cDNA by using the GoScript™ Reverse Transcription System (Promega, WI, USA), according to the manufacturer’s instructions. Using the gene-specific primers listed in Appendix A, qRT-PCR and RT-PCR were performed as follows: First hold at 94 °C for 5 min, followed by cycles at 94 °C for 30 s, 60 °C for 30 s, and 72 °C for 30 s and second hold at 72 °C for 5 min. The KAPA SYBR FAST qRT-PCR kit (KAPA Biosystem, Cape Town, South Africa) and CFX 96 thermal cycler (Bio-Rad, CA, USA) were used for qRT-PCR. To normalize gene expression and compare RNA levels, *PpACT2* (*P.patens.003G133100*) was used [73].

### 4.4. Subcellular Localization Using Fluorescent Reporter Proteins

To generate the eYFP-fused PpGPAT9 in the pPZP212 binary vector [74] harboring the *CaMV 35S* promoter, eYFP and rbs S3′ terminator, amplified cDNA fragment of *PpGPAT9* (primer sets are listed in Appendix A) was inserted using the restriction enzymes *Bam*HI and *Sal*I (referred to as peYFP:PpGPAT9). The recombinant binary vector was transformed into *Agrobacterium* strain GV3101 by using the freeze-thaw method. *Agrobacterium* cells harboring the *peYFP:PpGPAT9* construct was incubated up to OD_600_ = 0.8. The composition of the infiltration medium was as follows: 10 mM MES (pH 5.7), 10 mM MgCl_2_, and 0.2 mM acetosyringone. *Agrobacterium* cells in the infiltration media were inoculated using a 1 mL syringe. After 2 days, the fluorescent signal was observed using a TCS SP5 AOBS/tandem laser confocal scanning microscope (Leica, Wetzlar, Germany). *Agrobacterium* cells harboring *pBrFAD2-1:mRFP* construct were co-infiltrated to confirm ER localization [45].

### 4.5. Ectopic Expression and Transformation

To generate transgenic Arabidopsis ectopically expressing *PpGPAT9*, the amplified cDNA fragment of *PpGPAT9* (primer sets are listed in Appendix A) was inserted into the pBA002 binary vector [75] harboring the *CaMV 35S* promoter, 6x MYC, and NOS terminator. The recombinant binary vector was transformed into *Agrobacterium* strain GV3101 by using the freeze-thaw method. The *Agrobacterium* cells were incubated up to OD_600_ = 0.8 on YEP media containing rifampicin (50 µg/mL) and spectinomycin (50 µg/mL), and then they were collected and resuspended in transformation media (5% sucrose and 0.05% Silwet L-77). Arabidopsis transformation was performed using the floral-drop method [76].

### 4.6. Protein Extraction and Western Blotting

Crude proteins from 10-day-old seedlings of the wild type and transgenic Arabidopsis (T_2_) ectopically expressing *PpGPAT9* were extracted using extraction buffer (125 mM Tris-HCl buffer [pH 6.8], 0.2 M DTT, 4% SDS, 10% glycerol, 2 mM PMSF, 5 µg/mL aprotinin, 3 µg/mL pepstatin A, and 3 µg/mL leupeptin). Approximately 50 µg proteins were used for SDS-PAGE (10% gel) analysis (Mini-Protein Tetra System, Bio-Rad) and immunoblot assay. The proteins were transferred onto PVDF membrane and fixed with 5% formaldehyde and washed with pure water. The membrane was treated with blocking solution (5% skim milk in 1× TBST) on the rocker at room temperature for 1 h and then incubated with MYC antibody (clone 9E10, Millipore) on the rocker at 4 °C overnight. The membrane was incubated with anti-mouse IgG (GE Healthcare) at room temperature for 1 h. To detect the proteins, enhanced chemiluminescence reaction (ECL, Thermo Scientific, MA, USA) was performed using Amersham Imager 600 (GE Healthcare Life Sciences, MA, USA).

### 4.7. Microscopic Analysis of Seeds

Dry seeds (approximately 100 seeds) were treated with 8 M NaOH solution for 5 min to obtain soft tissues. The seeds were washed with distilled water and then cleared in Visikol (https://visikol.com/) for more than 10 h until they were submerged and transparent. The seeds were spread on a microscope slide with a few drops of Visikol solution. Then, the coverslip was placed on the microscope slide and gently pushed using forceps to separate the embryo from the seed coat. The microscopic image analyses were performed using LEICA ICC50 HD with Leica Application Suite (LAS) software (Leica, Wetzlar, Germany).

### 4.8. Fatty Acid Analysis

The dry seeds (100 seeds) and plant tissue powder (about 10 mg) were incubated in methylation solution (500 µL toluene, 1 mL of 5% sulfuric acid in methanol, and 500 µg glyceryl triheptadecanate as the standard) at 90 to 95 °C for 1.5 h. The FAMEs were extracted by adding 1.5 mL of 0.9% NaCl and sequential hexane. The supernatant was evaporated with nitrogen gas, and the FAMEs were concentrated with 100 µL hexane. The FAMEs were analyzed with GC-2010 (Shimadzu, Japan) and DB-23 column (30 mm × 0.25 mm, 0.25 µm film thickness; J&W Scientific, Folsom. CA, USA) as follows: temperature hold at 190 °C for 10 min, followed by 190 °C to 230 °C with 5 °C increase per min, and hold at 230 °C for 10 min.

### 4.9. Transient Expression in N. benthamiana Leaves

*Agrobacterium* harboring the pBA002 construct, *35S pro:AtWRI1*, and *35SP:6xMYC-PpGPAT9* construct or *P19* construct was incubated up to OD_600_ = 0.8 and diluted to OD_600_ = 0.2 by using infiltration media in the following combinations: pBA002 and *P19* or *AtWRI1* and *P19* or *35SP:6xMYC-PpGPAT9* and *P19*. To visualize the oil bodies in *N. benthamiana* leaves, the leaf disks were stained with Nile red solution (10 mg/mL in 0.1 M Tris-HCl buffer [pH 8]; Sigma-N3013) at room temperature for 30 min after 5 days of infiltration. The fluorescence was observed using the TCS SP5 AOBS/Tandem confocal laser scanning microscope (Leica Microsystems, Wetzlar, Germany). The emission wavelength was 615 nm, and the excitation wavelength was 560–572 nm.

## Figures and Tables

**Figure 1 plants-08-00284-f001:**
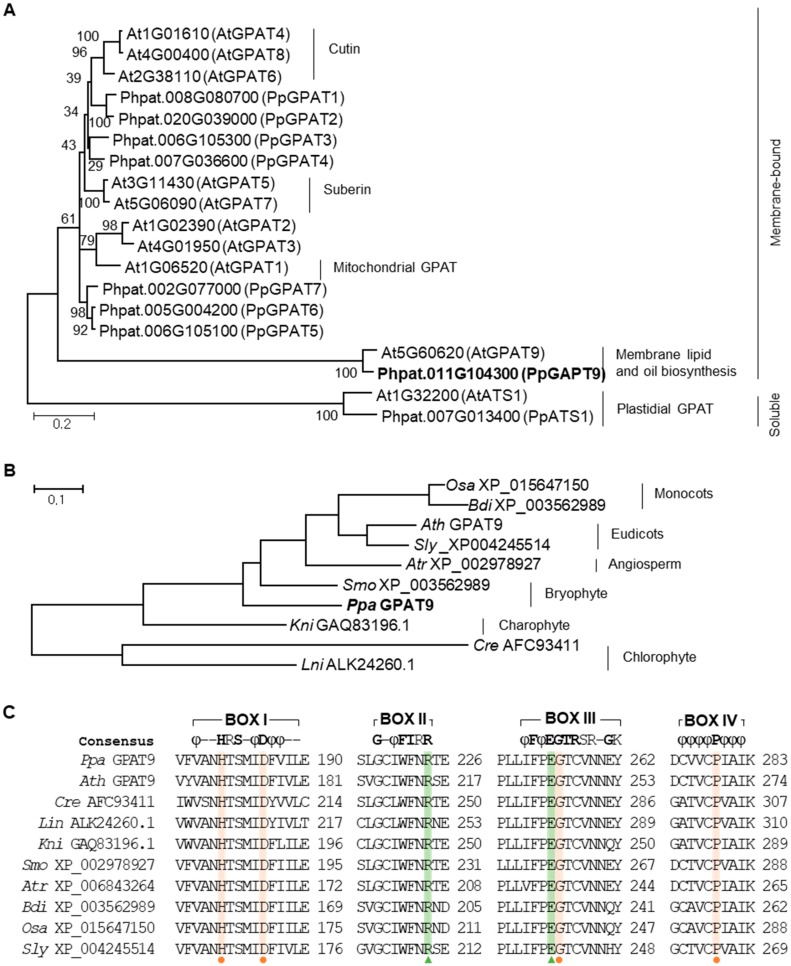
Phylogenetic trees for PpGPATs and AtGPATs (**A**) and PpGPAT9 and GPAT9s from other organisms (**B**), and conserved acyltransferase motifs (**C**) of GPAT9s. (**A**) The phylogenetic trees were constructed using the MEGA6 program and amino acid sequences of *A. thaliana* and *P. patens* GPATs obtained from the Phytozome v11.0 database (https://phytozome.jgi.doe.gov/) by using BLAST with a cutoff e-value of 1 × 10^−5^. (**B**) Homologs of PpGPAT9 were isolated from the genome database by using BLAST with a cutoff e-value of 1e-5 in Phytozome v12.1 (https://www.phytozome.net), NCBI (https://ncbi.nlm.nih.gov/), and UniProt (http://www.uniprot.org). Phylogenetic analysis was conducted using the Maximum Likelihood method in MEGA6 with multiple amino acid sequences. PpGPAT9 indicated in bold was used in this study. (**C**) Amino acid sequence alignment of *GPAT9* isoforms from *P. patens* and other organisms by using CLUSTALW. Four conserved acyltransferase motifs (BOX I–IV) are shown in colored boxes, according to Lewin et al. [44]. The amino acid residues indicated using orange circles in BOX I, BOX III, and BOX IV are required for GPAT catalysis. Hydrophobic amino acids are represented by “φ.” The following amino acids are considered hydrophobic: V, I, L, F, W, Y, and M. The amino acid residues indicated using green triangles in BOX II and BOX III are required for binding glycerol-3-phosphate. *Ppa* GPAT9 (*Physcomitrella patens* XP_024389305), *Ath* GPAT9 (*Arabidopsis thaliana* NP_568925), *Cre* AFC93411 (*Chlamydomonas reinhardtii*), *Lin* ALK24260 (*Lobosphaera incisa*), *Kni* GAQ83196.1 (*Klebsormidium nitens*), *Smo* XP_002978927 (*Selaginella moellendorffii*), *Atr* XP_006843264 (*Amborella trichopoda*), *Bdi* XP_003562989 (*Brachypodium distachyon*), *Osa* XP_015647150 (*Oryza sativa*), and *Sly* XP_004245514 (*Solanum lycopersicum*) are GenBank accession numbers.

**Figure 2 plants-08-00284-f002:**
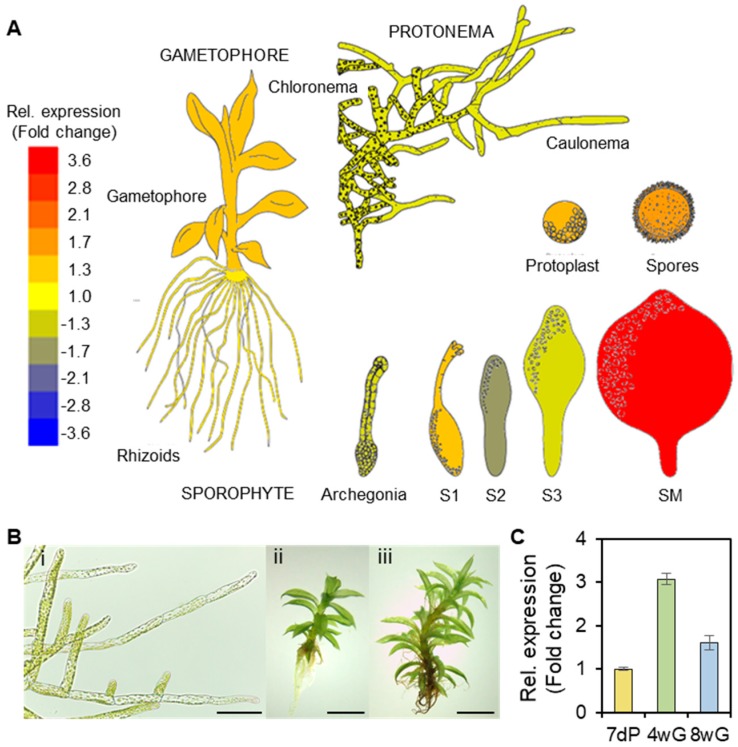
Expression pattern of *PpGPAT9* in *P. patens*. (**A**) The expression of *PpGPAT9* (*Pp1s138_27V6.1*, *Phpat.011G104300*) relative to *PpActin2* (*Pp1s198_157V6.1*, *Phpat.003G133100*). Images and values were obtained from *P. patens* eFP-Browser (http://bar.utoronto.ca/efp_physcomitrella/cgi-bin/efpWeb.cgi). Three- to 4-week-old gametophore; sporophyte 1 (S1), comprising sporophytes collected 5–6 days after fertilization (AF); sporophyte 2 (S2), 9–11 days AF; sporophyte 3 (S3), 18–20 days AF; and sporophyte M (SM), 28–33 days AF. (**B**) Morphology of *P. patens*. (i), Seven-day-old filamentous protonema, Bar = 0.4 mm. (ii), Four-week-old gametophore, Bar = 2 mm. (iii), Eight-week-old gametophore, Bar = 2 mm. (**C**) Quantitative RT-PCR analysis of *PpGPAT9*. Total RNA was isolated from 7-day-old protonema (7dP), 4-week-old gametophores (4wG), and 8-week-old gametophores (8wG). *PpActin2* was used as the reference gene to assess RNA quality and quantity. Each value is the mean ± SD of five independent measurements.

**Figure 3 plants-08-00284-f003:**
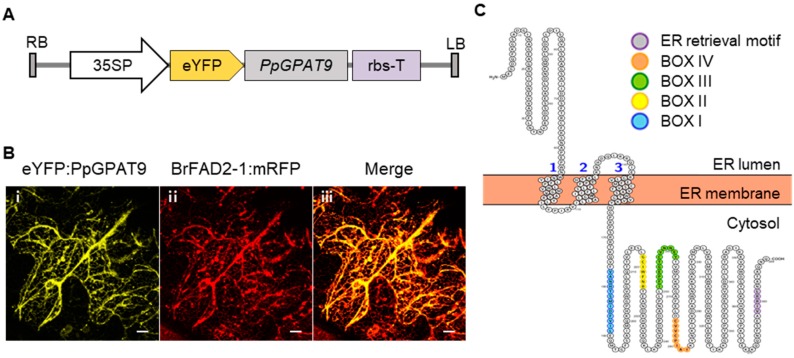
Subcellular localization of PpGPAT9. (A and B) Subcellular localization of eYFP:PpGPAT9 in *N. benthamiana* leaf epidermal cells. (**A**) Binary vector construct harboring eYFP:PpGPAT9. 35SP, cauliflower mosaic virus 35S promoter. eYFP, enhanced yellow fluorescent protein; LB, left border; RB, right border; rbs-T, terminator of ribulose-1,5-bisphosphate carboxylase and oxygenase small subunit from pea (*Pisum sativum*). (**B**) Two fluorescent signals in *N. benthamiana* leaf. *Agrobacterium tumefaciens* cells harboring eYFP:PpGPAT9 and BrFAD2-1:mRFP were co-infiltrated into *N. benthamiana* leaf epidermal cells. The fluorescent signals were visualized under a confocal laser-scanning microscope at 48 h after infiltration. (i) Yellow fluorescent signals from eYFP:PpGPAT9. (ii) Red fluorescent signals from BrFAD2-1:mRFP. (iii) Merged image of eYFP:PpGPAT9 with BrFAD2-1:mRFP. Bars = 10 μm. (**C**) Topology of PpGPAT9. The topology of PpGPAT9 was analyzed using the open-source tool for visualization of proteoforms (http://topcons.net/; http://wlab.ethz.ch/protter/start/) [47]. Computational analysis showed that PpGPAT9 harbors an N-terminal region and two non-cytoplasmic loops, three transmembrane regions, and one cytoplasmic loop and long C-terminal cytoplasmic regions. Acyltransferase motifs (BOX I–IV) and ER retrieval motif of long C-terminal cytoplasmic regions are shown in blue, yellow, green, orange, and purple, respectively. The numbers indicate the counts of each transmembrane domain.

**Figure 4 plants-08-00284-f004:**
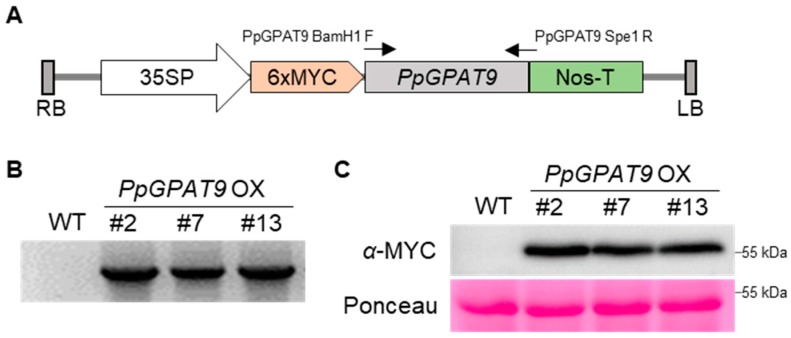
Generation of transgenic *A. thaliana* plants ectopically expressing *PpGPAT9* under the control of the *CaMV 35S* promoter. (**A**) Schematic diagram of the 35SP:6xMYC-PpGPAT9 construct. 35SP, Cauliflower mosaic virus 35S promoter; LB, left border; Nos-T, nopaline synthase terminator; RB, right border. The arrows indicate forward and reverse primer sites. (**B**) Genomic DNA (gDNA) PCR. The gDNA was isolated from the wild type (WT) and 3-week-old transgenic leaves ectopically expressing *PpGPAT9* (#2, #7, and #13) of the T_1_ generation. (**C**) Immunoblot analysis. Total proteins were extracted from 10-day-old seedlings of WT and transgenic lines ectopically expressing *PpGPAT9* of the T_2_ generation. Anti-myc antibodies were used to detect chemiluminescence (upper image) after 1 min of exposure. Protein quality and quantity were confirmed using Ponceau S (bottom image).

**Figure 5 plants-08-00284-f005:**
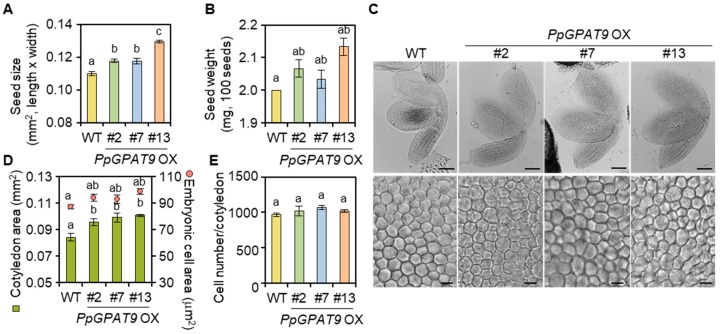
Microscopic analysis of seeds of the WT and transgenic Arabidopsis ectopically expressing *PpGPAT9* (#2, #7, and #13). (**A**) Length and width of seeds of the WT and transgenic lines (#2, #7, and #13). The data were statistically examined using ANOVA with Tukey’s test at 95% confidence interval. (**B**) Seed weight per 100 mature dried seeds of the WT and transgenic lines (#2, #7, and #13). The data were statistically examined using ANOVA with Tukey’s test at 95% confidence interval. (**C**) Images of mature embryos (upper image) and embryonic cells in the central region of the cotyledon (bottom image) obtained from dried seeds of the WT and transgenic lines (#2, #7, and #13). Bars = 0.1 mm (upper image). Bars = 0.01 mm (bottom image). (**D**) Cotyledon size and epidermal cell size of embryonic cells of the WT and transgenic lines (#2, #7, and #13). Each value is the mean ± SE of three independent measurements. The data were statistically analyzed using ANOVA with Tukey’s test at 95% confidence interval. (**E**) Epidermal cell number of adaxial cotyledons in the WT and transgenic lines (#2, #7, and #13). Each value is the mean ± SE of three independent measurements. The data were statistically analyzed using ANOVA with Tukey’s test at 95% confidence interval.

**Figure 6 plants-08-00284-f006:**
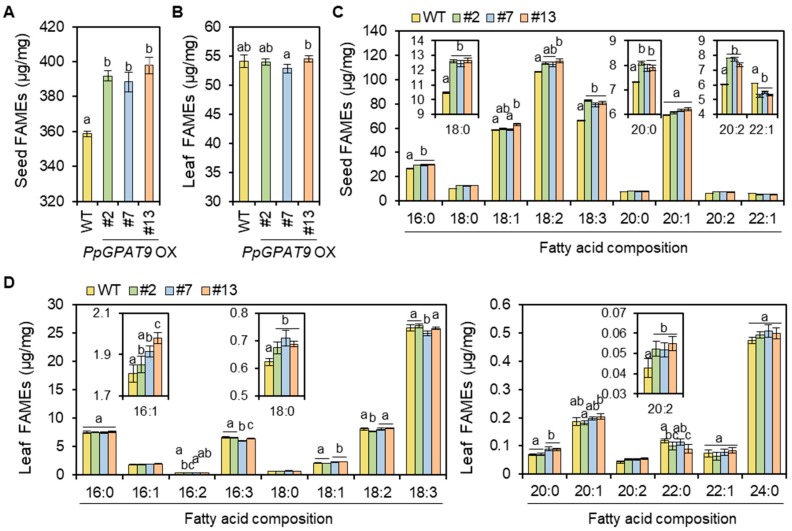
Fatty acid content and composition in the seeds (**A** and **C**) and leaves (**B** and **D**) of the WT and transgenic lines ectopically expressing *PpGPAT9* (#2, #7, and #13). Hundred dry seeds and 4-week-old leaves were used to extract the fatty acids. FAMEs were determined using GC-FID. Error bars indicate mean ± SE of three independent measurements. The data were statistically examined using ANOVA with Tukey’s test at 95% confidence interval. The percentage values indicate the portion of increased fatty acids when compared with the wild type.

**Figure 7 plants-08-00284-f007:**
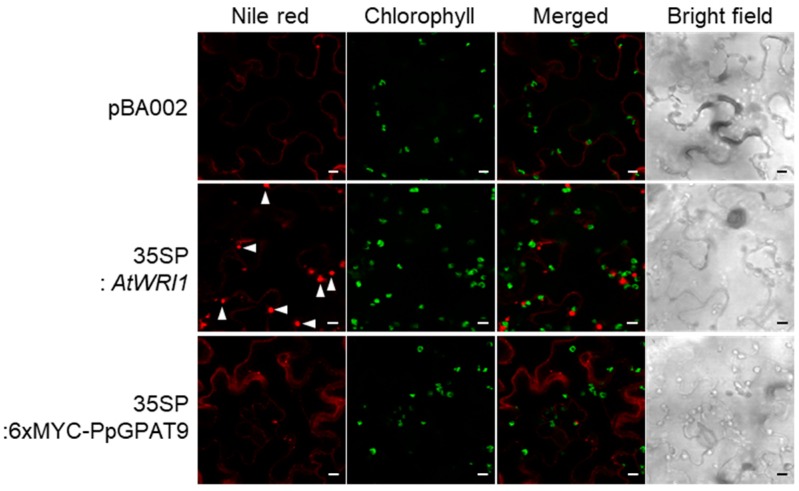
Transient expression of *PpGPAT9* in *N. benthamiana* leaves. *Agrobacterium* harboring *pBA002* (for the control); *AtWRI1* or *PpGPAT9* was infiltrated into *N. benthamiana* leaves, and the leaf disks were stained with Nile red solution. The fluorescent signals were visualized using confocal laser scanning microscopy. The arrowheads indicate oil bodies. Bars = 20 μm.

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
