# Peer review of "Functional Characterization of Physcomitrella patens Glycerol-3-Phosphate Acyltransferase 9 and an Increase in Seed Oil Content in Arabidopsis by Its Ectopic Expression"

_plants, 2019, doi:10.3390/plants8080284_

Round 1

Reviewer 1 Report

The manuscript by Mi Chung Suh Lab reported on the characterization of P patens GPAT9. They identified PpGPAT9, examined the expression in P patens, found ER localization in transient assay in Tobacco leaf, and produced multiple lines of overexpressors in Arabidopsis to investigate its function in vivo for lipid production and embryo development. GPAT9 is an important enzyme in glycerolipid metabolism, and P patens is an emerging model whose lipid metabolism is not well documented. So, this work is novel and is important in understanding the glycerolipid metabolism in different model species. I have a few suggestions and comments to improve the manuscript;

Major issues:

Despite a large increase in FAMEs in the seeds of PpGPAt9 OX (Fig. 6), observation of lipid droplets did not show a marked difference (Fig. 7). Where do these FAMEs come from? The authors should quantify TAG contents in these 3 transgenic lines overexpressing PpGPAT9 (#2, #7, and #13) and WT to address this point. This is necessary to support the statement in L276-277 and L351. Introduction section was a bit unclear to me with regard to the main scope of this work. Did the authors study GPAT9 because of interest in GPAT (or Kennedy pathway) in P patens, or TAG accumulation in P patens? I guess it is the latter case as the authors did not analyze polar glycerolipids (phospholipids and galactolipids) in the overexpression lines. In this case, the 2nd paragraph can be shortened by citing a recent review covering this part of the metabolism (e.g. Nakamura, Trends Plant Sci 2017 22(12):1027-1040) and instead the authors should give a clear justification to focus on GPAT9 in oil production. Discussion section can be better organized in my opinion. The first and last paragraphs are overlapping in summarizing the main finding, which is actually unnecessary to be in this section. It should be at least combined and the last paragraph should give a concluding remarks and a bit of future perspectives. I would suggest the authors to elaborate more on the possible mechanism behind an enlarged size of embryos in the overexpressors. There may be something more to be discussed based on the cited papers [46,55,58,61,62].

Minor editorial comments:

L15: “glycerol” should be “glycerol 3-phosphate”

L16: “, which produces high levels of very-long-chain PUFAs” in what organism?

L31: “from mainly “ should be “mainly from”?

L40: “glycerol 3-phosphate” (strike “-“ out between “glycerol” and “3-phosphate”)

L41: reference 10 (a review article on animal GPATs) should be better replaced by a couple of original papers on GPAT1 (Zou Jitao Lab) and GPAT9 (Phil Bates Lab [30])

L44: “PAP” is a more commonly used abbreviation than “PP” for PA phosphatase

L46: “FAD-4” should be “FAD4” (no “-“). The other FADs should be corrected likewise.

L51: Here, PDAT-mediated TAG biosynthesis pathway should be mentioned as well.

L88: “PpGPAT1” should be “PpGPAT9”?

L104-106: Since the phylogenetic trees are based on the amino acid sequences, “GPAT” should not be italicized here.

L174: “eukaryotic glycerolipid synthesis” should be “eukaryotic pathway of glycerolipid synthesis”

L351: “increased TAG levels” I found total FAME contents in Fig. 6 but not TAG data in the manuscript.

Author Response

Response to Reviewer 1 Comments

Comments and Suggestions for Authors

The manuscript by Mi Chung Suh Lab reported on the characterization of P. patens GPAT9. They identified PpGPAT9, examined the expression in P patens, found ER localization in transient assay in Tobacco leaf, and produced multiple lines of overexpressors in Arabidopsis to investigate its function in vivo for lipid production and embryo development. GPAT9 is an important enzyme in glycerolipid metabolism, and P patens is an emerging model whose lipid metabolism is not well documented. So, this work is novel and is important in understanding the glycerolipid metabolism in different model species. I have a few suggestions and comments to improve the manuscript;

Major issues:

Comment 1: Despite a large increase in FAMEs in the seeds of PpGPAt9 OX (Fig. 6), observation of lipid droplets did not show a marked difference (Fig. 7). Where do these FAMEs come from? The authors should quantify TAG contents in these 3 transgenic lines overexpressing PpGPAT9 (#2, #7, and #13) and WT to address this point. This is necessary to support the statement in L276-277 and L351.

Response 1: Based on the previous reports that the level of triacylglycerol (TAG) DW–1 is ~500-fold higher in Arabidopsis seeds compared with leaves (350 μg mg versus 0.6 μg mg DW–1; Li et al., 2006; Yang and Ohlrogge, 2009; Sanjaya et al., 2013), fatty acid methylesters (FAMEs) mainly came from triacylglycerols in seeds (Fig. 6A and C), and mainly from membrane lipids in leaves (Fig. 6B and D). Therefore TAG content was measured from seeds (Fig. 6A and C). Because there was no significant changes in the levels of total FAMEs in leaves, we did not measure the levels of TAG from leaves. Fig. 6B and D is supported by Fig. 7. The big differences in the levels of TAG between seeds and leaves may result from different mechanism underlying the turnover of TAG in seeds and leaves (Tjellstrom et al., 2015).

Li M Welti R Wang X . 2006. Quantitative profiling of Arabidopsis polar glycerolipids in response to phosphorus starvation. Roles of phospholipases Dζ1 and Dζ2 in phosphatidylcholine hydrolysis and digalactosyldiacylglycerol accumulation in phosphorus-starved plants. Plant Physiology 142, 750–761.

Yang Z Ohlrogge JB . 2009. Turnover of fatty acids during natural senescence of Arabidopsis, Brachypodium, and switchgrass and in Arabidopsis beta-oxidation mutants. Plant Physiology 150, 1981–1989.

Sanjaya Miller R Durrett TP et al.   2013. Altered lipid composition and enhanced nutritional value of Arabidopsis leaves following introduction of an algal diacylglycerol acyltransferase 2. The Plant Cell 25, 677–693.

Tjellstrom H, Strawsine M, Ohlrogge J. 2015. Tracking synthesis and turnover of triacylglycerol in leaves. Journal of Experimental Botany 66, 1453–1461

Comment 2: Introduction section was a bit unclear to me with regard to the main scope of this work. Did the authors study GPAT9 because of interest in GPAT (or Kennedy pathway) in P patens, or TAG accumulation in P patens? I guess it is the latter case as the authors did not analyze polar glycerolipids (phospholipids and galactolipids) in the overexpression lines. In this case, the 2nd paragraph can be shortened by citing a recent review covering this part of the metabolism (e.g. Nakamura, Trends Plant Sci 2017 22(12):1027-1040) and instead the authors should give a clear justification to focus on GPAT9 in oil production.

Response 2: We do understand the reviewer’ concern. Thus, the first sentence in the 2nd paragraph (line 40) in the revised manuscript was modified to clarify the sentence. And we appreciate for reviewer’ suggestion to recommend a recent review (Nakamura, Trends Plant Sci 2017 22(12):1027-1040) The reference [14] was included in the revised manuscript.

Comment 3: Discussion section can be better organized in my opinion. The first and last paragraphs are overlapping in summarizing the main finding, which is actually unnecessary to be in this section. It should be at least combined and the last paragraph should give a concluding remarks and a bit of future perspectives. I would suggest the authors to elaborate more on the possible mechanism behind an enlarged size of embryos in the overexpressors. There may be something more to be discussed based on the cited papers [46,55,58,61,62].

Response 3: Thank you for reviewer’ suggestion. The final paragraph was shortened in the revised manuscript (Lines 381-383). And the possible mechanism underlying the enlarged size of embryos in the overexpression lines was included in the discussion section (Lines 358-361) in the revised manuscript..

Comment 4: Minor editorial comments:

Minor 1: L15: “glycerol” should be “glycerol 3-phosphate”

Minor 2: L16: “, which produces high levels of very-long-chain PUFAs” in what organism?

Minor 3: L31: “from mainly “should be “mainly from”?

Minor 4: L40: “glycerol 3-phosphate” (strike “-“ out between “glycerol” and “3-phosphate”)

Minor 5: L41: reference 10 (a review article on animal GPATs) should be better replaced by a couple of original papers on GPAT1 (Zou Jitao Lab) and GPAT9 (Phil Bates Lab [30])

Minor 6: L44: “PAP” is a more commonly used abbreviation than “PP” for PA phosphatase

Minor 7: L46: “FAD-4” should be “FAD4” (no “-“). The other FADs should be corrected likewise.

Minor 8: L51: Here, PDAT-mediated TAG biosynthesis pathway should be mentioned as well.

Minor 9: L88: “PpGPAT1” should be “PpGPAT9”?

Minor 10: L104-106: Since the phylogenetic trees are based on the amino acid sequences, “GPAT” should not be italicized here.

Minor 11: L174: “eukaryotic glycerolipid synthesis” should be “eukaryotic pathway of glycerolipid synthesis”

Response 4-1: Thank you for your comments. Based on the reviewer’s comments, all points were edited or some information was included in the revised manuscript, except minor 12. Please see the revised manuscript.

Minor 12: L351: “increased TAG levels” I found total FAME contents in Fig. 6, but not TAG data in the manuscript.

Response 4-2: Please see our response 1.

Reviewer 2 Report

This appears to be a generally well organized and well written manuscript. 

Line 71                  delete “existing on earth”

Figure 1 A and B               

                                Suggest that the authors amend the Figure 1 legend to include what each of the “other

organisms” is. For example – from what organism was Osa XP_015647150 obtained?

Line 272                change “alternations” to changes              

Line 288                delete “was”

Line 307                consider changing “Arabidopsis” to A. thaliana

Line 316                contained within the ………

Check to ensure that Arabidopsis,  Agrobacterium, etc. are appropriately italicized (or not) per the formatting guidelines of the journal.  Same for gene designations.    

Author Response

Response to Reviewer 2 Comments

Comments and Suggestions for Authors

This appears to be a generally well organized and well written manuscript. 

Comment 1

#1. Line 71   delete “existing on earth”

#2. Figure 1 A and B Suggest that the authors amend the Figure 1 legend to include what each of the “other organisms” is. For example – from what organism was Osa XP_015647150 obtained?

#3, Line 272  change “alternations” to changes  

Response 1-1: Thank you for your comments. Based on the reviewer’s comments, all points were edited or some information was included in the revised manuscript, except minor #4, #5, and #6. Please see the revised manuscript.

#4, Line 288  delete “was”

#5, Line 307  consider changing “Arabidopsis” to A. thaliana

Response 1-2: We are thinking that “was” should be there and “Arabidopsis” might be O.K.

#6, Line 316  contained within the ………

Response 1-3: We could not find the words in Line 316 of the original manuscript.

Comment 2: Check to ensure that Arabidopsis, Agrobacterium, etc. are appropriately italicized (or not) per the formatting guidelines of the journal.  Same for gene designations.    

Response 2: Thank you for your comments. Based on the reviewer’s comments, all points were checked.

Reviewer 3 Report

In this study, Yang et al identified PpGPAT9 in P. patens, expressed it in A. thaliana, and observed its effect on mainly seed oil content.

Each data is clear and seems to be good, however, this version of the manuscript is lacking novelty not only fundamental aspect but also applied science points of view to publish in this journal. 

I left some comments to improve your article below:

Major points

1) If the authors want to increase the seed oil contents in A. thaliana (higher plants), AtGPAT9 overexpression is the better way since Singer et al. (2016) have already reported that AtGPAT9 overexpression resulted in increases of seed oil content, seed weight/area, and 18:1 and 22:0 fatty acid composition in the seed.

More recently, Fukuda et al (2018) have also reported that GPAT9-type GPAT overexpression enhances drastically TAG accumulation (more than 56-fold), and its fatty acid compositions, especially 18:2 and 20:2, were increased in microalga.    

If the authors want to demonstrate GPAT9-type GPAT is involved in storage oil production, the point is not new at all and not be attractive for the reader of this journal. If the oil content by PpGPAT9 expression can be drastically improved in higher plants such as A. thaliana, they can sate that PpGAPT9 is a "novel genetic resource" to enhance storage oil yields in the seed. However, the effect is extremely minor to conclude that and there is also a possibility that the changes observed in this study are caused by the secondary effect of P. patens derived gene expression in A. thaliana.

2) The authors must be careful using the terminology of “overexpression” in this article. You can't say "PpGPAT9 overexpression" because PpGPAT9 is exogenous gene; Figure 4 is just showing exogenous PpGPAT9 is expressing, but not showing overexpressing in the cells.   

3) They measured only FAMEs, but they have to measure what's kind of lipids were accumulated in the PpGPAT expressing strain. Further, they need to define clearly "storage oils". Does it mean TAGs? 

Minor points

1) Line 15: glycerol must be G3P

2)Figure 1A: The number of proteins (only two organisms) is too small to indicate the phylogenetic relationships among the GPAT proteins. 

3) Figure 4B: The authors must indicate molecular size in the ponceau stained membrane.

4) Figure 5C: PpGPAT9 OE must be changed to PpGPAT9 OX

5) Lines 351 -352, In the current study, increased TAG levels in

PpGPAT9 OX seeds were observed to be proportional to the increased cotyledon size:

Which data is indicting "increased TAG levels in PpGPAT9 OX seeds"?

6) Lines 361-366: They should indicate Reference(s) for these data.  

7) As I mentioned Major point 1, GPAT9-type GPAT has been also characterized in microalgae [Fukuda et al. (2018)]. So, the authors must pay attention to the previous finding(s) and also discuss the functions in the part of DISCUSSION.

Author Response

Response to Reviewer 3 Comments

Comments and Suggestions for Authors

In this study, Yang et al identified PpGPAT9 in P. patens, expressed it in A. thaliana, and observed its effect on mainly seed oil content.

Each data is clear and seems to be good, however, this version of the manuscript is lacking novelty not only fundamental aspect but also applied science points of view to publish in this journal. 

I left some comments to improve your article below:

Major points

Comment 1: If the authors want to increase the seed oil contents in A. thaliana (higher plants), AtGPAT9 overexpression is the better way since Singer et al. (2016) have already reported that AtGPAT9 overexpression resulted in increases of seed oil content, seed weight/area, and 18:1 and 22:0 fatty acid composition in the seed.

More recently, Fukuda et al (2018) have also reported that GPAT9-type GPAT overexpression enhances drastically TAG accumulation (more than 56-fold), and its fatty acid compositions, especially 18:2 and 20:2, were increased in microalga. If the authors want to demonstrate GPAT9-type GPAT is involved in storage oil production, the point is not new at all and not be attractive for the reader of this journal. If the oil content by PpGPAT9 expression can be drastically improved in higher plants such as A. thaliana, they can sate that PpGAPT9 is a "novel genetic resource" to enhance storage oil yields in the seed. However, the effect is extremely minor to conclude that and there is also a possibility that the changes observed in this study are caused by the secondary effect of P. patens derived gene expression in A. thaliana.

Response 1: Thank you for your critical comments. Thus we changed the title “Overexpression of Physcomitrella patens glycerol-3-phosphate acyltransferase 9 led to an increase in seed oil content in Arabidopsis “ to “Functional characterization of Physcomitrella patens glycerol-3-phosphate acyltransferase 9 and an increase in seed oil content in Arabidopsis by its ectopic expression” to clarify the significance of this manuscript. And Singer et al. (2016) and Fukuda et al (2018) papers [33, 34], which were recommended by reviewer 3, were included in the revised manuscript to inform what is different between this study and previous observations.

Comment 2: The authors must be careful using the terminology of “overexpression” in this article. You can't say "PpGPAT9 overexpression" because PpGPAT9 is exogenous gene; Figure 4 is just showing exogenous PpGPAT9 is expressing, but not showing overexpressing in the cells.   

Response 2: Thank you for your comment. ‘overexpression’ was changed to ‘ectopic expression’.

Comment 3: They measured only FAMEs, but they have to measure what's kind of lipids were accumulated in the PpGPAT expressing strain. Further, they need to define clearly "storage oils". Does it mean TAGs? 

Response 3: Based on the previous reports that the level of triacylglycerol (TAG) DW–1 is ~500-fold higher in Arabidopsis seeds compared with leaves (350 μg mg versus 0.6 μg mg DW–1; Li et al., 2006; Yang and Ohlrogge, 2009; Sanjaya et al., 2013), fatty acid methylesters (FAMEs) mainly came from triacylglycerols in seeds (Fig. 6A and C), and mainly from membrane lipids in leaves (Fig. 6B and D). The points were clarified in the revised manuscript (Line 260, Line276).

Li M Welti R Wang X . 2006. Quantitative profiling of Arabidopsis polar glycerolipids in response to phosphorus starvation. Roles of phospholipases Dζ1 and Dζ2 in phosphatidylcholine hydrolysis and digalactosyldiacylglycerol accumulation in phosphorus-starved plants. Plant Physiology 142, 750–761.

Yang Z Ohlrogge JB . 2009. Turnover of fatty acids during natural senescence of Arabidopsis, Brachypodium, and switchgrass and in Arabidopsis beta-oxidation mutants. Plant Physiology 150, 1981–1989.

Sanjaya Miller R Durrett TP et al.   2013. Altered lipid composition and enhanced nutritional value of Arabidopsis leaves following introduction of an algal diacylglycerol acyltransferase 2. The Plant Cell 25, 677–693.

Comment 4: Minor points

1) Line 15: glycerol must be G3P

3) Figure 4B: The authors must indicate molecular size in the ponceau stained membrane.

4) Figure 5C: PpGPAT9 OE must be changed to PpGPAT9 OX

6) Lines 361-366: They should indicate Reference(s) for these data.  

Response 4-1: All points were revised. Please see the revised manuscript.

2) Figure 1A: The number of proteins (only two organisms) is too small to indicate the phylogenetic relationships among the GPAT proteins. 

Response 4-2: Arabidopsis contains 10 GPAT isoforms and some of them were functionally identified. Thus we want to examine phylogenetic relationship of multiple GPAT isoforms from Arabidopsis and P. patens to intrigue initial clue of PpGPAT9 function.

5) Lines 351-352, In the current study, increased TAG levels in PpGPAT9 OX seeds were observed to be proportional to the increased cotyledon size: Which data is indicting "increased TAG levels in PpGPAT9 OX seeds"?

Response 4-2: Fatty acid methylesters (FAMEs) mainly came from triacylglycerols in seeds (Fig. 6A and C). Please see our response 3.

7) As I mentioned Major point 1, GPAT9-type GPAT has been also characterized in microalgae [Fukuda et al. (2018)]. So, the authors must pay attention to the previous finding(s) and also discuss the functions in the part of DISCUSSION.

Response 4-3: Thank you for your suggestion. The conclusion in the discussion section was modified in the revised manuscript (Lines 381-383). And please see our response 1.

Reviewer 4 Report

This manuscript reports on the function of moss GPAT9 in glycerolipid metabolism. GPAT9 is an essential enzyme that catalyzes the first reaction step of de novo phospholipid biosynthesis pathway in the ER. Lipid metabolism in moss species is poorly described so far. Here, the authors used P patens and Arabidopsis to perform a through characterization of newly isolated PpGPAT9 including in silico phylogenetic analysis, expression study in P patens, subcellular localization study, and a transgenic approach to overexpress GPAT9 in Arabidopsis and assess its effect on glycerolipid metabolism and oil contents in seeds. I found this is a very substantial work with solid set of data. Results are presented in a succinct manner, and the data are reasonably and carefully interpreted. Overall, the work has a good value in the relevant research fields.

I have only a couple of minor suggestions to improve the manuscript before it can be acceptable for publication;

Throughout the manuscript, it was unclear to see if the focus of this work is to characterize GPAT9 or to develop an approach to increase oil contents. Apparently, the manuscript has a value in the former aspect over the latter aspect in which the increase in oil content is not surprisingly high. Thus, to avoid any misleading underestimation of the value of this work, the authors are encouraged to emphasize on the scope of this work towards the characterization of moss GPAT9. Discussion section was a bit lengthy and contain many repetitive descriptions of the result and somewhat irrelevant discussion. In my view, it can be condensed so the main points are comprehensively covered and discussed. The authors mention oil contents following the overexpression of GPAT9, but the data in Fig. 6 are the total FAMEs. While it is obviously derived from TAGs as SE is not enriched with FAs, the authors should be noted with this issue and revise the related statements in the text in a more accurate way.

Author Response

Response to Reviewer 4 Comments

This manuscript reports on the function of moss GPAT9 in glycerolipid metabolism. GPAT9 is an essential enzyme that catalyzes the first reaction step of de novo phospholipid biosynthesis pathway in the ER. Lipid metabolism in moss species is poorly described so far. Here, the authors used P patens and Arabidopsis to perform a through characterization of newly isolated PpGPAT9 including in silico phylogenetic analysis, expression study in P patens, subcellular localization study, and a transgenic approach to overexpress GPAT9 in Arabidopsis and assess its effect on glycerolipid metabolism and oil contents in seeds. I found this is a very substantial work with solid set of data. Results are presented in a succinct manner, and the data are reasonably and carefully interpreted. Overall, the work has a good value in the relevant research fields.

I have only a couple of minor suggestions to improve the manuscript before it can be acceptable for publication;

Comment 1: Throughout the manuscript, it was unclear to see if the focus of this work is to characterize GPAT9 or to develop an approach to increase oil contents. Apparently, the manuscript has a value in the former aspect over the latter aspect in which the increase in oil content is not surprisingly high. Thus, to avoid any misleading underestimation of the value of this work, the authors are encouraged to emphasize on the scope of this work towards the characterization of moss GPAT9.

Response 1: Thank you for your critical comments. Thus we changed the title “Overexpression of Physcomitrella patens glycerol-3-phosphate acyltransferase 9 led to an increase in seed oil content in Arabidopsis “ to “Functional characterization of Physcomitrella patens glycerol-3-phosphate acyltransferase 9 and an increase in seed oil content in Arabidopsis by its ectopic expression” to clarify the significance of this manuscript.

Comment 2: Discussion section was a bit lengthy and contain many repetitive descriptions of the result and somewhat irrelevant discussion. In my view, it can be condensed so the main points are comprehensively covered and discussed.

Response 2: According to the reviewer’ comments, discussion part was condensed in the revised manuscript.

Comment 3: The authors mention oil contents following the overexpression of GPAT9, but the data in Fig. 6 are the total FAMEs. While it is obviously derived from TAGs as SE is not enriched with FAs, the authors should be noted with this issue and revise the related statements in the text in a more accurate way.

Response 3: The part was clarified in the revised manuscript (Line 260 and Line276) and the relevant references [48-50] were included.

Round 2

Reviewer 3 Report

The authors improved the points what I concerned.

However, one point has not been revised completely.

----

Response 2: Thank you for your comment. ‘overexpression’ was changed to ‘ectopic expression’.

---

The authors are still using "overexpression" in many places in the revised manuscript.

All ‘overexpression’ must be changed to ‘ectopic expression’.

Also, if the authors will use OX in the strain name, they need to add some explanation about that and/or definition in the text.

Author Response

Reviewer's comments

The authors improved the points what I concerned.

However, one point has not been revised completely.

The authors are still using "overexpression" in many places in the revised manuscript.

All ‘overexpression’ must be changed to ‘ectopic expression’.

Also, if the authors will use OX in the strain name, they need to add some explanation about that and/or definition in the text.

Response:

Thanks.

We changed "overexpressing" to "ectopically expressing" in the revised manuscript.